# Does Vaccination Protect against Human Papillomavirus-Related Cancers? Preliminary Findings from the United States National Health and Nutrition Examination Survey (2011–2018)

**DOI:** 10.3390/vaccines10122113

**Published:** 2022-12-10

**Authors:** Alpamys Issanov, Mohammad Ehsanul Karim, Gulzhanat Aimagambetova, Trevor J. B. Dummer

**Affiliations:** 1School of Population and Public Health, Faculty of Medicine, University of British Columbia, Vancouver, BC V6T 1Z3, Canada; 2Centre for Health Evaluation and Outcome Sciences, St. Paul’s Hospital, Vancouver, BC V6Z IY6, Canada; 3Department of Biomedical Sciences, School of Medicine, Nazarbayev University, Astana 010000, Kazakhstan

**Keywords:** human papillomavirus, HPV vaccine, HPV-related cancers, NHANES

## Abstract

Most oropharyngeal and anogenital cancers are caused by human papillomavirus (HPV). Although HPV vaccines showed high efficacy against oropharyngeal and anogenital HPV infections, and cancer precursors in randomized clinical trials, there are limited data on the effectiveness of HPV vaccination against HPV-related cancers. We aimed to evaluate the association of HPV vaccination with HPV-related cancers among a nationally representative sample of United States adults, aged 20–59 years. In a cross-sectional study combining four cycles from the National Health and Nutrition Examination Survey, from 2011 through 2018, we used a survey-weighted logistic regression model, propensity score matching and multiple imputations by chained equations to explore the association of HPV vaccination with HPV-related cancers. Among 9891 participants, we did not find an association of HPV vaccination with HPV-related cancers (adjusted OR = 0.58, 95% CI 0.19; 1.75). Despite no statistically significant association between HPV vaccination and HPV-related cancers, our study findings suggest that HPV-vaccinated adults might have lower odds of developing HPV-related cancers than those who were not vaccinated. Given the importance of determining the impact of vaccination on HPV-related cancers, there is a need to conduct future research by linking cancer registry data with vaccination records, to obtain more robust results.

## 1. Introduction

Oropharyngeal cancers, including cancers of the lip, oral cavity, pharynx, and larynx, and anogenital cancers, such as cervical, vaginal, vulvar, penile and anal cancers, are common cancers worldwide [1]. The American Cancer Society estimated 54,000 and 40,810 new cases, and 11,230 and 9610 new deaths related to oropharyngeal and anogenital cancers, respectively, in the U.S. in 2022 [2]. While risk factors for these cancers are age [2], ethnicity [2], smoking [3], alcohol consumption [3], environmental pollutants [4], and dietary factors [5], a sexually transmitted human papillomavirus (HPV) infection plays a major role in their development [6]. It has been shown that HPV is causally associated with almost all cervical cancers [7], and it is also the cause of some genital cancers, collectively called HPV-related cancers [7]. Seven high-risk HPV types (16, 18, 31, 33, 45, 52 and 58) are causally associated with 90% of cervical, 75% vaginal, 69% vulvar, 63% penile, and 70% oropharyngeal cancer cases [8]. There is also some evidence of potential associations of HPV with other genital cancers (e.g., prostate cancer, testicular cancer) [9,10].

There are highly efficacious HPV vaccines against the high-risk HPV infection types, recommended at the age of 9–14 for girls and boys [11]. The Advisory Committee on Immunization Practices also recommends catch-up HPV vaccination for all individuals up to 26 years old, while HPV vaccination for adults aged 27–45 years should be based on clinical decision-making [12,13]. In randomized clinical trials, HPV vaccines showed high efficacy in reducing rates of cervical cancer precursors [14] and decreasing rates of oral and anogenital HPV infections [15]. However, there are limited data on the effectiveness of HPV vaccination against HPV-related cancers [15,16]. Multiple studies examined the post-licensure effect of HPV vaccine on HPV infection rates [17,18,19]. Several cross-sectional analyses of the National Health and Nutrition Examination Survey (NHANES) data have found that young people who self-reported being previously HPV vaccinated had considerably lower rates of oral HPV infections [17]. Additionally, descriptive epidemiological studies showed a decrease in incidence rates of cervical and vaginal cancers in the United States (U.S.) from 1999 to 2015 [18]. However, only two studies investigated the effects of HPV vaccination on HPV-related cancers [20,21]. Linking to cancer registry data, one study reported high efficacy of HPV vaccination against oropharyngeal cancers among cohorts who were previously enrolled in HPV vaccine trials, although the detected number of cancer cases was small [20]. The second study found lower incidence rates of invasive cervical cancer among HPV-vaccinated cohorts compared to unvaccinated, utilizing the Sweden population registry [21].

Given the limited literature and high vaccine hesitancy rates in eligible U.S. adolescents [22,23], there is a need to understand the impact of HPV vaccination on HPV-related cancer rates and to inform policymakers, adolescents eligible for vaccination and parents considering vaccinating their children about the effectiveness of HPV vaccines against the HPV-related cancers. The findings will also be useful in promoting HPV immunization campaigns and supporting parents and adolescents in decision-making about HPV vaccination. In this study, we utilized the NHANES cycles from 2011 through 2018 to investigate the following research question: “Is HPV vaccination associated with a lower risk of HPV-related cancers in U.S. adults aged 20–59?”. We hypothesized that the risks of HPV-related cancers would be lower among the HPV vaccinated population than a non-HPV-vaccinated population. This study performed logistic regression modeling appropriate for complex surveys to examine the relationship between HPV vaccination and HPV-related cancers in the general adult U.S. population, utilizing nationally representative data.

## 2. Materials and Methods

### 2.1. Study Design and Data Source

In the cross-sectional study, we analyzed four cycles (2011–2018) from NHANES, a continuous survey designed to assess health and dietary status of the non-institutionalized U.S. population [24]. Bi-annually, a nationally representative sample is selected using a complex multistage probability sampling design [25]. The survey details and sample design have been described elsewhere [26]. The ethical approval for the study was received under item 7.10.3 in the University of British Columbia’s Policy 89 on studies involving human participants [27] and Article 2.2 in the Tri-Council Policy Statement Ethical Conduct for Research Involving Humans [28].

### 2.2. Study Population

Overall, 39,156 participants responded to health and dietary questions from the four NHANES cycles [29]. 24,222 participants younger than 20 years or older than 59 years were excluded because they were not eligible to respond to questions about cancer status and HPV vaccination history. Responses “Refused”, “Don’t know” or “Not stated” were treated as missing values. We excluded all participants with at least one missing value in any variable. The final complete case sample included 9891 participants. For more details about the selection process of the study sample, see Figure 1. 

### 2.3. Study Variables

Two survey questions were used to develop the binary outcome variable: a self-reported diagnosis of HPV-related cancers. The first question was: “Have you ever been told by a doctor or other health professional that you had cancer or a malignancy of any kind?” [29]. The second question, asked to those who responded “yes” to the first question, was: “What kind was it?”. Participants who reported being diagnosed with larynx, windpipe, mouth, tongue, lip or cervical cancers (anal, penile, vulvar, and vaginal cancers were not self-reported) were categorized as HPV-related cancers, while those who were not diagnosed with cancer or were diagnosed with other cancer types were categorized as non-HPV-related cancers. 

The primary exposure variable was HPV vaccination history. It was directly obtained from the survey question: “Have you ever received one or more doses of the HPV vaccine?” [29]. Those participants who reported receiving at least one dose were categorized as HPV vaccinated, while those who did not receive any dose were categorized as not HPV vaccinated.

Based on previous literature and epidemiological reasoning, the following covariates were considered as potential confounders of the association of HPV vaccination history with HNGC: age [3,30,31], ethnicity [30,31], education [3,22,31], household income [22], whether participants were born in the U.S. or not [31], and NHANES cycle. Potential risk factors for HNGC included marital status, any history of consuming 4–5 alcohol drinks daily, having smoked at least 100 cigarettes over their lifetime, self-reported diet, a history of being overweight, a history of diabetes, moderate or vigorous physical activity at work, and routine access to healthcare services “when sick or need advice about health” [31]. The assumed causal relationships are depicted in Figure 2.

### 2.4. Statistical Analysis

#### 2.4.1. Primary Analysis

To account for the complex sampling design, survey weights, strata and clusters were used in analyzing data. Survey weights were divided by 4 to account for combining four NHANES cycles. In the descriptive analysis, categorical variables were presented as sample frequencies, while percentages were presented as weighted estimates for the target population. Associations between categorical variables and the outcome variable were tested using Rao-Scott chi-square test for complex survey design studies [32]. A survey-weighted multiple logistic regression model was built to test the association of HPV vaccination history with HPV-related cancers using the complete case dataset. We used a directed acyclic graph with the backdoor criterion to select a minimum sufficient adjustment set for the model [33]. Previous literature suggested that age, ethnicity, education, household income, and whether individuals were born in the U.S. were associated with HPV vaccination and with HPV-related cancers [3,31]. We also assumed that timing would be associated with both the exposure and the outcome; thus, in addition to the aforementioned confounders, NHANES cycle was also added to the model. After retaining the variables blocking the backdoor paths in the model, potential risk factors for the outcome were selected using the backward elimination approach, based on the Akaike Information Criterion [34]. We used the Archer-Lemeshow test for design-based regression models [35] and the Area Under the Receiver Operating Characteristics Curve [36] to test the model’s goodness-of-fit. A *p*-value less than 0.05 was considered a statistically significant finding.

#### 2.4.2. Secondary Analysis (Propensity Score Matching)

Additionally, we performed propensity score-matched analysis to test the robustness of the results; this is considered a secondary confounding adjustment approach to the conventional logistic regression analysis when there are fewer events per confounder [37]. Propensity score (PS) matching was applied using a 1:1 nearest-neighbor method (without replacement), with a caliper width of 0.2 of the standard deviation of the logit of the propensity score [38]. We estimated PS using a logistic regression model including confounding variables, risk factors for the outcome (as shown in Figure 2), and survey features as recommended by Dugoff et al. [39]. We examined covariate balance between matched HPV-vaccinated and non-HPV-vaccinated groups using standardized mean difference (SMD). SMD < 0.1 was indicative of adequate covariate balance between the matched groups [40,41]. A crude survey-weighted logistic regression for the outcome was conducted to estimate the average treatment effect on the treated in the matched subsample.

#### 2.4.3. Sensitivity Analysis

While the complete case analysis was more a conservative approach to address missing or invalid responses in covariates, the analysis actually excluded more than 4000 participants. We could not assume that the data was missing completely at random because distributions of sociodemographic characteristics of the study participants—including sex, ethnicity, whether individuals were born in the U.S. or not, income, smoking history, and physical activity—differed between the complete case dataset and the dataset with missing values (Table 1). Assuming that the missing data can be explained by the observed data, i.e., missing at random assumption [42], and attempting to increase a statistical power of the analysis, we imputed missing data for covariate variables using the “multiple imputations then deletion” approach [43]. The missing values were imputed by chained equations using the “mice” package [44]. We used 20 imputations. Binary and polytomous logistic regression models were applied to impute missing values for binary and multi-level categorical variables, respectively. We used all variables to predict missing values, including the complex survey design features [45]. Survey-weighted outcome regression analysis was performed by forcing confounding variables in the imputation models, while the risk factors for the outcome were selected using AIC-based backward elimination by “stacking” all multiply imputed datasets [46]. Estimates from the imputation models were pooled using Rubin’s rules [44]. All statistical analyses were performed using the R-software version 4.1.1 (Vienna, Austria) [47].

## 3. Results

### 3.1. Sample Characteristics and Univariate Analysis

Overall, 0.7% (72/9891) participants reported being diagnosed with HPV-related cancers in the complete case sample (Table 1). Sixy-eight out of 72 HPV-related cancers were cervical cancers. The prevalence rate of HPV-related cancers decreased in the last two NHANES cycles (Figure 3). On the other hand, HPV vaccination prevalence rates were increasing from 5.5% to 13.3% (Rao-Scott chi-square test, *p* < 0.05) between 2011 and 2018. 76.8% of all HPV vaccinated participants were in the age group 20–29 years (Appendix A). HPV vaccination rates were comparable between all participants and those participants who were not diagnosed with HPV-related cancers (9.4% and 9.4%, respectively), while participants who were diagnosed with HPV-related cancers had the lowest HPV vaccination rate (2.6%, *p* < 0.01) (Table 1). There were statistically significantly differences between participants who were diagnosed with HPV-related cancers and those who were not diagnosed with HPV-related cancers in age (*p* = 0.001), sex (*p* < 0.001), education (*p* < 0.01), ethnicity (*p* = 0.01), household income (*p* = 0.001), having smoked at least 100 cigarettes over their lifetime *p* < 0.001), and routine access to healthcare services (*p* < 0.01).

### 3.2. Primary Analysis

After AIC-based stepwise backward elimination, the survey-weighted final multiple logistic regression model, assessing the association of HPV vaccination history with HPV-related cancers, was adjusted for age, education, ethnicity, marital status, whether individuals were born in the U.S. or not, income, having smoked at least 100 cigarettes, moderate or vigorous physical activity at work, a history of consuming 4/5 alcohol drinks daily, a history of being overweight, routine access to healthcare services and NHANES cycle (Table 2). In this model, there was no statistically significant association between HPV vaccination history and HPV-related cancers (adjusted OR = 0.58, 95% CI 0.19; 1.75).

### 3.3. Secondary Analysis

891 HPV-vaccinated participants were matched with 891 non-HPV-vaccinated participants using PS matching approach. Covariates between the matched groups were adequately balanced–SMDs were <0.1 for all covariates (Figure 4). In the survey-weighted simple logistic regression in the matched dataset (Table 2), no statistically significant association of HPV vaccination history with HPV-related cancers was observed (crude OR = 0.40, 95% CI 0.10; 1.69). 

### 3.4. Sensitivity Analysis

25.7% of data a history of consuming 4/5 alcohol drinks every day, 6.6% of data for household income and less than 1% of data for other covariates were missing (see the Table 2 footnotes for details). The proportion of HPV-related cancers was lower (0.9%, not shown in Table 1) in the missing data than in the complete data. There were also differences in sex, education, ethnicity, location of birth, household income, having ever smoked at least 100 cigarettes over a lifetime, a history of being overweight, and moderate or vigorous physical activity at work between the complete dataset and the missing dataset. After applying the “multiple imputations then deletion” method, each imputed dataset contained 13,993 individuals. Pooled estimates from imputed models adjusting for age, education, ethnicity, marital status, whether they born in the U.S. or not, income, having ever smoked at least 100 cigarettes, moderate or vigorous physical activity at work, ever drank 4/5 alcohol drinks every day, a history of being overweight, routine access to healthcare services and NHANES cycle revealed no statistically significant association of HPV vaccination history with HPV-related cancers (adjusted OR = 0.47, 95% CI 0.15; 1.45; see in Table 2).

## 4. Discussion

In this nationally representative study, the primary analysis showed no association of HPV vaccination with HPV-related cancers. Although the relationship was not statistically significant, participants who were immunized with at least one dose of a HPV vaccine had 42% odds of HPV-related cancers (adjusted OR = 0.58, 95% CI: 0.19; 1.75) than those who were not HPV vaccinated. When conducting the propensity score-matched analysis and the analysis on multiple imputed datasets to assess the association between vaccination and HPV-related cancers, we also did not find a statistically significant association. However, a similar trend toward reduction of odds among those who were vaccinated (60% and 53%, respectively) was observed in the alternative analysis and the sensitivity analysis. The alternative and sensitivity analyses also provided narrower confidence intervals for OR, as they were able to better deal with the small number of HPV-related cancer cases per covariate in the model and potentially increase statistical power by imputing missing data.

A similar result was reported in a population-based study from Sweden [21]. Utilizing the Swedish Total Population Register, researchers compared incidence rates of HPV-associated cervical cancer between vaccinated and unvaccinated populations from 2006 to 2017. They found that the incidence rate ratio (IRR) was 0.51 (95% CI 0.32; 0.82) for vaccinated relative to unvaccinated populations; after adjusting for more covariates, the association became even stronger: IRR = 0.37 (95% CI, 0.21; 0.57) [21]. Our findings are also consistent with results from a Finnish follow-up study [20]. Linking women who were previously enrolled in HPV vaccine clinical trials to the Finish Cancer Registry, researchers were able to compare HPV-associated cancer rates between vaccinated and unvaccinated cohorts [20]. They observed ten cancer cases caused by HPV (eight cervical cancers, one oropharyngeal cancer and one vulvar cancer) among previously unvaccinated cohorts, while no cancer cases were observed in vaccinated cohorts. Thus, the study reported an HPV vaccine efficacy estimate of 100% (95% CI 16; 100) [20]. Similarly, a study from England reported relative incidence rate reductions in cervical cancer among women who were offered the vaccine as follows: between age 16–18 years: 34% (95% CI 25%–41%); between age 14–16 years: 62% (95% CI 52%–71%); and between age 12–13 years: 87% (95% CI 72%–94%), in comparison to unvaccinated women [48]. Another study from Denmark found a statistically significant reduction in incidence rate of cervical cancer in women who were vaccinated at age 16 years or younger—86% (95% CI 47%; 96%)—while failing to find any reduction at ages 17 years or older [49]. Lastly, the comprehensive HPV vaccination program in Australia is expected to halve cervical cancer rates by 2035 [50].

We noticed a reduction in HPV-related cancer cases in the last two NHANES cycles, relative to the previous 2011–2012 and 2013–2014 cycles. Similarly, a trend analysis of HPV-associated cancers between 1999 and 2015 using data from cancer registries in the U.S. showed a decline in incidence rates of cervical cancer in all age groups (1.6% decrease per year), vaginal cancer among <40 years old (2.8% decrease per year) and anal cancer among <40-year-old men (2.9% decrease per year) [18]. In Australia, since the introduction of the National HPV Vaccination Programme—first among girls in 2007, then among boys in 2013—HPV vaccine type prevalence and incidence of genital cancer precursors have declined among vaccinated women [51]. 

We also found a significant increase in HPV vaccination rates from 2011 through 2018. Similarly, other studies have shown an increase in HPV coverage among adolescents [52,53] and adults aged 19–26 years in the U.S. [54]. As the efficacy and effectiveness of the HPV vaccine against HPV prevalence were previously reported [55,56], nationally representative studies reported a decline of HPV prevalence in the U.S. Results from the NHANES 2003–2010 cycles showed a substantial reduction in HPV vaccine type prevalence by 56% decrease among young females [56]. Another NHANES study compared HPV vaccine type prevalence between the pre-vaccination era (2003–2006) to post-vaccination era (2007–2014) [55]. The study results found that there was a 71% reduction in HPV prevalence among 14- to 19-year-old females and a 61% reduction among 20- to 24-year-old females between pre- and post-vaccination eras [55]. 

Although the efficacy and effectiveness of the HPV vaccine against HPV prevalence, HPV-associated anogenital warts and cervical cancer precursors is well supported by the evidence [16,17,19,56,57,58], our results are important for understanding and extending knowledge about the HPV vaccine effectiveness against HPV-related cancers, evidence of which is currently lacking. While several high-income countries (e.g., Norway, Italy and Denmark) [59,60,61] and a few middle-income countries (e.g., Uzbekistan) [62] have reached and maintained high HPV vaccination coverage, the U.S. is experiencing challenges to maintain an adequate vaccination rate [30,63]. On the other hand, only less than a quarter of low-income and less than third of lower-middle-income countries had introduced the HPV vaccination [64]. Given HPV vaccination rates are relatively low nationally [30,63] and internationally [65] and overall hesitation to introduce HPV vaccine in low- and middle-income countries [64], the study findings can be used by policymakers and health professionals to further encourage HPV vaccination uptake among adolescents aged 11–15 years and to support decision-making about HPV vaccination among adults eligible to be vaccinated and parents considering immunizing their children.

### Strengths and Limitations

The study has several strengths. Since we used a nationally representative sample, the study findings are generalizable to the adult non-institutionalized U.S. population. We also conducted secondary and the sensitivity analyses to check the robustness of our study findings. We found similar results across all analyses, suggesting the robustness of the results.

There are study limitations that need to be acknowledged. First, in our study, we do not know whether HPV-related cancers were caused by HPV infection, as the outcome data were collected based on self-reporting. Since most cases were cervical cancer (68 out of 72), it is unlikely that other factors might have contributed to their development. Second, differential misclassification bias might have been present. Other anogenital cancers, including vulvar, penile, vaginal, and anal cancers, were not separately reported in the survey, and they might have been classified together under the “Other cancers” group and not included in the outcome. Not including them in the study outcome might have weakened the association by pulling the effect estimate towards the null. 

Next, since the vaccination status was also based on self-reporting, we cannot exclude the existence of the recall bias, as participants might mistakenly report their HPV vaccination history. However, the recall bias might be limited, since participants were specifically asked whether they received a vaccine against HPV under brand names Cervarix, Gardasil or Gardasil 9 [29]. 

Since one-dose vaccination might not provide full protection against HPV infections, it might then be inappropriate to compare at least one-dose vaccinated versus unvaccinated. Given the small number of HPV-related cancer cases in HPV vaccinated group, it was numerically challenging for us to estimate the effects of full and partial HPV vaccination in relation to non-HPV vaccination in this study. 

Due to the small number of events, we were also not able to assess potential effect modifications of covariates (e.g., sex, ethnicity) on the relationship. As HPV vaccination before the first sexual onset, at age 11–12 years, is generally recommended, the vaccine might have a lower effectiveness in the older individuals, as they might have already contracted the infection. We were not able to determine whether participants were vaccinated before the first sexual onset or not, since only a few participants responded to questions about sexual behavior. 

Another limitation is that non-response bias might be present in the study. More than half of all participants did not know their vaccination status or refused to respond. When we compared characteristics of participants with missing or invalid responses for HPV vaccination status to the complete case data, participants with missing HPV vaccination data had higher rates of HPV-related cancers and were likely from the NHANES 2017–2018 cycle. Additionally, the study timing and the observed rarity of events were the major limitations to finding a statistically significant association. In addition to a small number of HPV-related cancers (*n* = 72), most vaccinated participants were between the age of 20–39 years (92.5%). Since HPV-related cancer development may take 10–20 years after first contracting HPV, the majority of participants were not yet at risk of developing cancers, reducing the probability of detecting an association [66,67,68]. Thus, having a sufficient number of events and comparable age distributions between HPV-vaccinated and unvaccinated groups are desirable to be able to detect statistically significant findings. A statistically significant association was observed when we performed an additional analysis with an expanded outcome definition by including other potentially HPV-associated genital cancers (Appendix A). 

Lastly, we also acknowledge that the present study had insufficient statistical power to detect a statistically significant difference between the vaccinated and unvaccinated groups (Appendix A). Future studies with large sample sizes are warranted to assess a statistically significant reduction.

## 5. Conclusions

While the analyses found no association of HPV vaccination with HPV-related cancers, the study findings suggest that HPV vaccinated people tend to have lower odds of developing HPV-related cancers than unvaccinated people. Given the importance of determining the impact of the vaccination on HPV-related cancers, there is a need to conduct future research by linking cancer registry data with vaccination records to obtain more robust results. It is understood that the effect of HPV vaccination will be observed in the next coming decades; however, preliminary findings, such as this study, could be used as additional evidence to encourage vaccination among 11- to 15-year-old adolescents and eligible adults, to reduce HPV-related cancer rates and prevent HPV-related cancer cases in the future.

## Figures and Tables

**Figure 1 vaccines-10-02113-f001:**
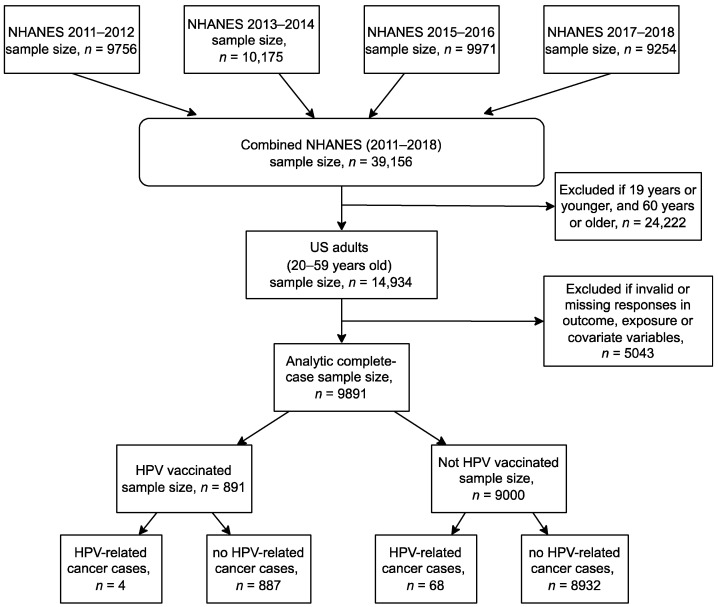
Flowchart describing the analytic complete case sample selection for a study investigating the relationship of HPV vaccination to HPV-related cancers using data from the National Health and Nutrition Examination Survey (NHANES) between 2011 through 2018. Abbreviations: U.S.: the United States; HPV: human papillomavirus.

**Figure 2 vaccines-10-02113-f002:**
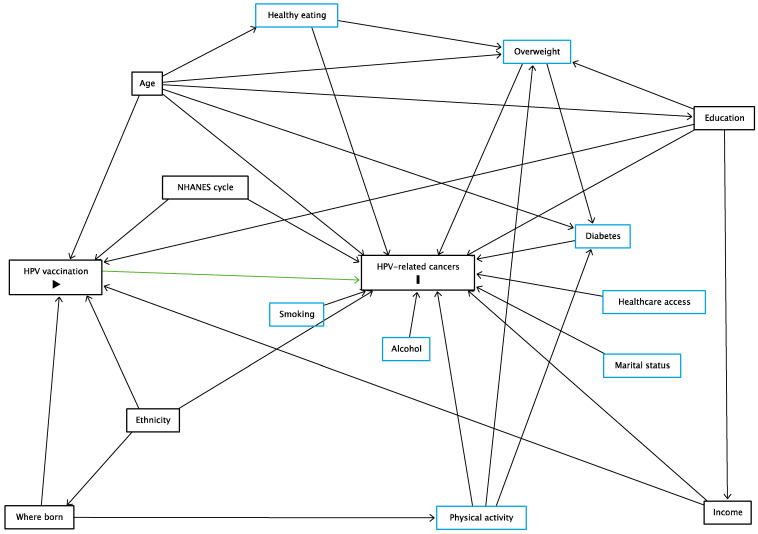
Directed acyclic graph examining the association of human papillomavirus (HPV) vaccination history with HPV-related cancers. Age, ethnicity, education, household income, and whether participants were born in the U.S. or not, and National Health and Nutrition Examination Survey (NHANES) cycle are considered as confounders; adjusting for these blocks all backdoor pathways (indicated in gray boxes). Marital status, any history of consuming 4–5 alcohol drinks daily (alcohol), having smoked at least 100 cigarettes over their lifetime (smoking), self-reported diet (healthy eating), a history of being overweight, a history of diabetes, moderate or vigorous physical activity at work, and routine access to healthcare services “when sick or need advice about health” (healthcare access) are assumed to be risk factors for HPV-related cancers (shown in blue, except “HPV-related cancers”, which is the outcome). The variable in green with sign “ 
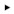
” is the primary exposure variable (HPV vaccination). The variable with “ 
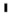
” sign is the outcome (HPV-related cancers).

**Figure 3 vaccines-10-02113-f003:**
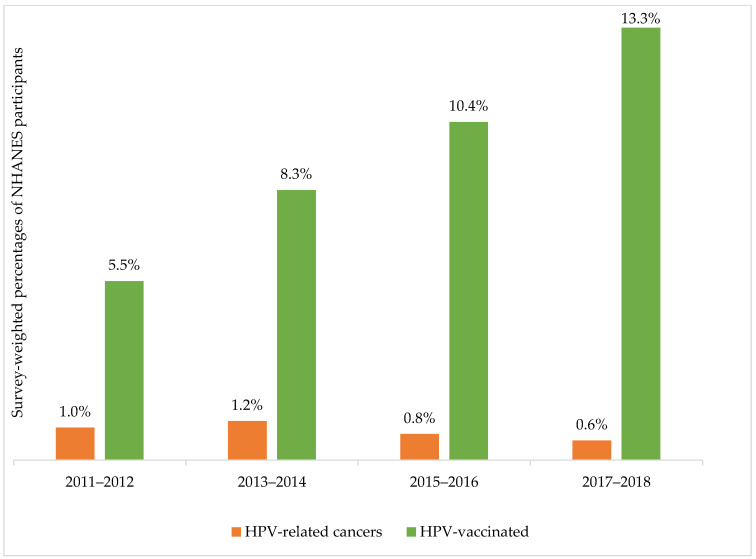
Prevalence of head, neck and genital cancers, prevalence of cervical cancer and HPV vaccination prevalence, stratified by the National Health and Nutrition Examination Survey (NHANES) cycles between 2011 through 2018.

**Figure 4 vaccines-10-02113-f004:**
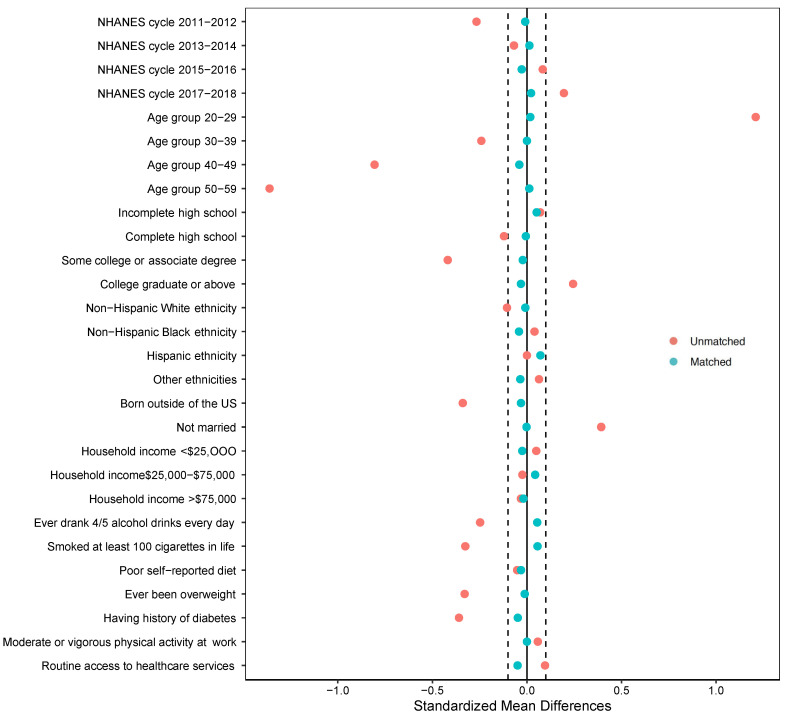
The covariate balance in the unmatched (complete case data) and matched datasets from a study investigating the association of human papillomavirus vaccination history with HPV-related cancers among U.S. adults aged 20–59 years in the National Health and Nutrition Examination Survey (NHANES), 2011–2018. Standardized mean differences for the unmatched dataset (*n* = 9891) are shown in red dots. Standardized mean differences for the matched dataset (*n* = 1782) are shown in blue dots.

**Table 1 vaccines-10-02113-t001:** Descriptive characteristics of U.S. adults aged 20–59 years, stratified by those who were diagnosed with HPV-related cancers and those who were not diagnosed with HPV-related cancers, using data from the National Health and Nutrition Examination Survey (NHANES), 2011–2018.

Variables	No HPV-Related Cancers (*n* = 9819)	HPV-Related Cancers (*n* = 72)	*p*-Value ^b^	All ^c^ (*n* = 9891)	Missing Data ^d^ (*n* = 4102)
	*n* (%) ^a^	*n* (%) ^a^		*n* (%) ^a^	*n* (%) ^a^
NHANES cycle			0.25		
2011–2012	2479 (24.9)	17 (28.2)		2496 (24.9)	1098 (25.9)
2013–2014	2682 (25.3)	26 (33.9)		2708 (25.3)	1033 (25.0)
2015–2016	2402 (24.5)	14 (20.3)		2416 (24.5)	1118 (26.8)
2017–2018	2256 (25.4)	15 (17.6)		2271 (25.3)	853 (22.4)
HPV vaccination history			0.01		
Not vaccinated	8932 (90.6)	68 (97.4)		9000 (90.6)	3779 (91.6)
Vaccinated	887 (9.4)	4 (2.6)		891 (9.4)	323 (8.4)
Age groups			0.001		
20–29 years	2358 (24.1)	5 (6.0)		2363 (24.0)	995 (25.5)
30–39 years	2484 (23.5)	19 (20.2)		2503 (23.5)	1046 (26.6)
40–49 years	2457 (25.2)	26 (40.2)		2483 (25.3)	1030 (23.2)
50–59 years	2520 (27.2)	22 (33.7)		2542 (27.2)	1031 (24.6)
Sex			<0.001		
Males	4942 (50.3)	3 (4.0)		4945 (49.8)	1626 (42.0)
Females	4877 (49.7)	69 (96.0)		4946 (50.2)	2476 (58.0)
Education			<0.01		
Did not complete high school	2712 (33.5)	6 (12.2)		2718 (33.3)	1019 (28.0)
Completed high school	2155 (21.4)	16 (21.2)		2171 (21.4)	870 (22.5)
Some college or associate degree	1648 (11.8)	17 (15.5)		1665 (11.8)	1015 (19.1)
College graduate or above	3304 (33.4)	33 (51.1)		3337 (33.5)	1192 (30.4)
Ethnicity			0.01		
Non-Hispanic White	2275 (15.5)	9 (6.1)		2284 (15.4)	1126 (22.7)
Non-Hispanic Black	2181 (11.2)	8 (3.5)		2189 (11.2)	984 (15.7)
Hispanic	3814 (65.1)	49 (82.6)		3863 (65.3)	919 (47.5)
Other ethnicities	1549 (8.1)	6 (7.8)		1555 (8.1)	1073 (14.0)
Where born			0.15		
In US	7262 (84.4)	65 (91.3)		7327 (84.5)	2123 (66.4)
Outside US	2557 (15.6)	7 (8.7)		2564 (15.5)	1975 (33.6)
Marital status			0.20		
Married or living with partner	5839 (63.2)	33 (54.6)		5872 (63.1)	2507 (61.3)
Not married	3980 (36.8)	39 (45.4)		4019 (36.9)	1590 (38.7)
Household income			0.001		
<$25,000	4253 (39.9)	33 (53.8)		4286 (40.1)	1420 (43.0)
$25,000-$75,000	3210 (43.4)	10 (19.9)		3220 (43.1)	867 (33.5)
>$75,000	2356 (16.7)	29 (26.3)		2385 (16.8)	937 (23.5)
Ever drank 4/5 alcohol drinks every day			0.58		
Yes	1539 (15.7)	15 (19.1)		1554 (15.7)	81 (15.5)
No	8280 (84.3)	57 (80.9)		8337 (84.3)	449 (84.5))
Smoked at least 100 cigarettes in life			<0.001		
Yes	4340 (44.5)	55 (72.8)		4395 (44.7)	1055 (28.7)
No	5479 (55.5)	17 (27.2)		5496 (55.3)	3041 (71.3)
Self-reported diet ^¶^			0.99		
Healthy	6423 (68.9)	43 (68.8)		6466 (68.9)	2885 (70.6)
Poor	3396 (31.1)	29 (31.2)		3425 (31.1)	1215 (29.4)
Ever been overweight ^†^			0.31		
Yes	5047 (52.5)	48 (60.6)		5095 (52.6)	1835 (46.3)
No	4772 (47.5)	24 (39.4)		4796 (47.4)	2264 (53.7))
History of diabetes			0.27		
Yes	947 (8.4)	10 (12.4)		957 (8.4)	390 (8.1)
No	8872 (91.6)	62 (87.6)		8934 (91.6)	3702 (91.9)
Moderate or vigorous physical activity at work			0.60		
Yes	4621 (49.8)	38 (54.1)		4659 (49.8)	1474 (39.1)
No	5198 (50.2)	34 (45.9)		5232 (50.2)	2622 (60.9)
Routine access to healthcare services ^‡^			<0.01		
Yes	7695 (80.1)	66 (93.2)		7761 (80.2)	3208 (78.2)
No	2124 (19.9)	6 (6.8)		2130 (19.8)	893 (21.8)

Abbreviations: HPV: human papillomavirus; U.S.: the United States of America. ^a^ Frequencies describe the study sample, while percentages represent survey weighted estimates for the US population. ^b^ The Rao-Scott chi-square test for complex survey design was used to calculate *p*-values. ^c^ Complete case analytic data. ^d^ Data with at least one missing value in any variable. ^¶^ To the question “How healthy is your overall diet?”, responses “excellent”, “very good” or “good” were categorized to “healthy”, while responses “fair” or “poor” were categorized as “poor”. **^†^** Overweight was defined as those who have ever told by a doctor that they were overweight or those whose body mass index was 25 or higher in the past 10 years. **^‡^** To the question “Is there a place that you usually go when you sick or you need advice about your health?”, responses “Yes” and “There is more than one place” were categorized as “Yes”, and “There is no place” as “No”.

**Table 2 vaccines-10-02113-t002:** Survey-weighted logistic regression models using complete case, propensity score-matched and multiply imputed datasets investigating the association of human papillomavirus vaccination history with HPV-related cancers among U.S. adults aged 20–59 years in the National Health and Nutrition Examination Survey (NHANES), 2011–2018.

Variables	aOR (95% CI) ^a^ Complete-Case Data ^b^	Crude OR (95% CI) PS-Matched Data ^c^	aOR (95% CI) ^d^ MI Data ^e^
HPV vaccination history			
Not vaccinated	Reference	Reference	Reference
Vaccinated	0.58 (0.19; 1.75)	0.40 (0.10; 1.69)	0.47 (0.15; 1.45)

Abbreviations: aOR: adjusted odds ratio; CI: confidence interval; OR: odds ratio, PS: propensity score; MI: multiple imputations; HPV: human papillomavirus. ^a^ The survey-weighted multivariable logistic regression model was adjusted for age, education, ethnicity, marital status, whether born in the U.S. or not, income, having smoked at least 100 cigarettes, moderate or vigorous physical activity at work, a history of consuming 4/5 alcohol drinks daily, a history of being overweight, routine access to healthcare services and NHANES cycle. The goodness-of-fit of the model using the Archer-Lemenshow test *p* < 0.05. The Area Under the Curve of the Receiver Operating Characteristics Curve was 0.81. ^b^ The sample size in complete-case data was 9891. ^c^ Propensity score-matched data included 891 vaccinated and 891 non-vaccinated participants. Propensity score matching was performed using a 1:1 nearest-neighbor method (without replacement) with a caliper width of 0.2 of the standard deviation of the logit of the propensity score. ^d^ The survey-weighted multivariable logistic regression model was adjusted for age, education, ethnicity, marital status, whether indvidiuals were born in the U.S. or not, income, having smoked at least 100 cigarettes, moderate or vigorous physical activity at work, a history of consuming 4/5 alcohol drinks every day, a history of being overweight, routine access to healthcare services and NHANES cycle. ^e^ Missing values for: a history of consuming 4/5 alcohol drinks every day (25.7%), income (6.6%) and other covariates with less 1% missing data (routine access to healthcare services, self-reported diet, a history of being overweight, marital status, having smoked at least 100 cigarettes over their lifetime, education, moderate or vigorous physical activity at work, location of birth, history of diabetes) were imputed using “multiple imputation then deletion” approach. 20 imputations were used. Each imputed dataset contained 13,993 observations. Estimates were pooled using the Rubin’s rules.

## Data Availability

Publicly available datasets were analyzed in this study. The National Health and Nutrition Examination Survey (NHANES) datasets can be found here: https://wwwn.cdc.gov/nchs/nhanes/ (accessed on 15 November 2022).

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
