# Peer review of "Does Vaccination Protect against Human Papillomavirus-Related Cancers? Preliminary Findings from the United States National Health and Nutrition Examination Survey (2011–2018)"

_vaccines, 2022, doi:10.3390/vaccines10122113_

Round 1

Reviewer 1 Report

Issano et al. have studied the risk of HPV-related cancer in 891 HPV-vaccinated and 9,000 non-vaccinated US adults 20-59 years of age. There were four HPV-related cancers (0.4%) among the vaccinated and 58 cases among the non-vaccinated (0.8%). The difference was not statistical significant.

Comments

1. Cervical cancer is most frequently diagnosed in women between the ages of 35 and 44 with the average age at diagnosis being 50. It rarely develops in women younger than 25. In the US, the incidence rate was 7.7 per 100,000 women per year in 2014-2018. You need more power than 891 HPV-vaccinated and 9,000 unvaccinated women 20-59 years old between 2011 and 2018 to see any statistically significant results. In the manuscript, about 50% of the participants were males, without any risk of cervical cancer. 

https://cancerstatisticscenter.cancer.org

2. Usually, cervical cancer and other HPV-related cancers develops over several years. It takes at least 10 years from HPV-infection to development of invasive cancer. 

It has been estimated that the mean calculated time from infection to diagnosis of CIN3+ is 9.4 years (SD 4.1 years) (Depuydt 2012) and progression from CIN3 to invasive cervical cancer takes 10–20 years, depending on genotype (Tjalma 2013). A modelling study by Burger et al. found that the median time from acquisition of HPV to cancer detection ranged from 17.5 years to 26.0 years (Burger 2020). It is not expected to find a significant reduction in the risk of HPV-related cancer the first 10-15 years after vaccination.

  1. Tjalma WA, Fiander A, Reich O, Powell N, Nowakowski AM, Kirschner B, et al. Differences in human papillomavirus type distribution in high-grade cervical intraepithelial neoplasia and invasive cervical cancer in Europe. Int J Cancer. 2013;132(4):854–67. https://doi.org/10.1002/ijc.27713.

  2. Depuydt CE, Criel AM, Benoy IH, Arbyn M, Vereecken AJ, Bogers JJ. Changes in type-specific human papillomavirus load predict progression to cervical cancer. J Cell Mol Med. 2012;16(12):3096–104. https://doi.org/10.1111/j.1582-4934.2012.01631.x.

  3. Burger EA, de Kok I, Groene E, et al. Estimating the natural history of cervical carcinogenesis using simulation models: a CISNET comparative analysis. J Natl Cancer Inst. 2020;112(9):955–63. https://doi.org/10.1093/jnci/djz227.

  4. 3. The Advisory Committee on Immunization Practices (ACIP)* routinely recommends HPV vaccination at age 11 or 12 years. Because HPV acquisition generally occurs soon after first sexual activity, vaccine effectiveness will be lower in older age groups because of priorinfections. Exposure to HPV decreases among older age groups.

Issano et al. studied US adults 20-59 years old. Only of few of these have been vaccinated before sexual onset.

4. Issano et al. have sited two studies showing reduction in HPV-related cancer among vaccinated women. Issano et al. should also include more studies.

In a study from England, the estimated relative reduction in cervical cancer rates by age at vaccine offer were 34% (95% CI 25–41) for age 16–18 years (school year 12–13), 62% (52–71) for age 14–16 years (school year 10–11), and 87% (72–94) for age 12–13 years (school year 8), compared with the reference unvaccinated cohort. 

Falcaro M, Castañon A, Ndlela B, Checchi M, Soldan K, Lopez-Bernal J, Elliss-Brookes L, Sasieni P. The effects of the national HPV vaccination programme in England, UK, on cervical cancer and grade 3 cervical intraepithelial neoplasia incidence: a register-based observational study. Lancet. 2021 Dec 4;398(10316):2084-2092. doi: 10.1016/S0140-6736(21)02178-4. Epub 2021 Nov 3. PMID: 34741816.

https://pubmed.ncbi.nlm.nih.gov/34741816/

In a study from Denmark including 867 689 women aged 17-30 years living in Denmark October 2006-December 2019, 36.3% were vaccinated at age 16 years and younger, and during follow-up, 19.3% and 2.3% were vaccinated at ages 17-19 years and 20-30 years, respectively. For women vaccinated at ages 16 years and younger or 17-19 years, the IRRs of cervical cancer were 0.14 (95% CI = 0.04 to 0.53) and 0.32 (95% CI = 0.08 to 1.28), respectively, compared with unvaccinated women.

Kjaer SK, Dehlendorff C, Belmonte F, Baandrup L. Real-World Effectiveness of Human Papillomavirus Vaccination Against Cervical Cancer. J Natl Cancer Inst. 2021 Oct 1;113(10):1329-1335. doi: 10.1093/jnci/djab080. PMID: 33876216; PMCID: PMC8486335.

https://pubmed.ncbi.nlm.nih.gov/33876216/

Recently, the International Papillomavirus Society (IPVS) issued a ‘call to action’ to health authorities to achieve elimination of cervical cancer as a public health problem. 1 In principle, cervical cancer rates can eventually be reduced to near-zero given the highly effective primary prevention (HPV vaccination) and secondary prevention (cervical screening) interventions now available.

Australia is poised to be the first country to approach cervical cancer elimination, since it has now fully implemented all these major prevention interventions. Australia was the first country to introduce a national publicly-funded HPV vaccination program in 2007, with a wide catch-up age range from 12 to 26 years. In 2013, Australia introduced vaccination for adolescent males, and in 2018 the next generation nonavalent vaccine was introduced, which protects against seven carcinogenic types which are associated with ~90% of cervical cancers. Multiple studies have documented the impact on health outcomes: the prevalence of vaccine-included type-specific infections in young women aged 25-35 years has now drop-ped by a factor of 10 (even in unvaccinated females, due to herd immunity), 5 the prevalence of anogenital warts has dropped substantially in both females and heterosexual males, 6 cervical precancerous abnormalities (CIN2/3) have now dropped by 41% nationally in women aged 2024 years, 7 and the rate of excisional treatment has now also fallen in young women. 8

In the intermediate term, cervical cancer rates are expected to halve (again) by 2035, and mortality rates should remain stable until about 2020, but then decline by 45% by 2035. These findings indicated that both HPV vaccination and primary HPV screening represent significant and timely steps in Australia’s journey towards elimination of cervical cancer.

https://www.hpvworld.com/articles/australia-on-track-to-be-the-first-country-to-achieve-cervical-cancer-elimination/

Minor revisions

Line 40, "High-risk HPV types – HPV 6, 11, 16, 18 and other types – are causally associated with 90% of cervical" => "Seven high-risk HPV types (16, 18, 31, 33, 45, 52 and 58) – are causally associated with 90% of cervical"

(HPV type 6 and 11 are low-risk HPV-types causing genital warts and not HPV-related cancers. HPV type 16 and 18 are causing 70% of all cases of cervical cancer. The 9-valent HPV-vaccine covers the 7 hrHPV-types causing 90% of all cases of cervical cancer).

Line 44-45, "There are highly efficacious HPV vaccines against the high-risk HPV infection types, recommended at the age of 9-14 for girls and boys"

Females between 9 and 45 years of age can be vaccinated with Cervarix, Gardasil or Gardasil 9 to prevent cervical cancer and precancerous cervical changes. Gardasil and Gardasil 9 may also prevent vaginal, vulvar and anal cancers and their precancers, as well as anogenital warts. 

In Canada, about one-third of HPV-related cancers occur in males. Gardasil and Gardasil 9 are available for boys and young men between the ages of 9 and 26 to prevent anal cancer, its precancer and anogenital warts.

https://cancer.ca/en/cancer-information/reduce-your-risk/get-vaccinated/human-papillomavirus-hpv

The Advisory Committee on Immunization Practices (ACIP)* routinely recommends HPV vaccination at age 11 or 12 years; vaccination can be given starting at age 9 years. Catch-up vaccination has been recommended since 2006 for females through age 26 years, and since 2011 for males through age 21 years and certain special populations through age 26 years. This report updates ACIP catch-up HPV vaccination recommendations and guidance published in 2014, 2015, and 2016 (13). Routine recommendations for vaccination of adolescents have not changed. In June 2019, ACIP recommended catch-up HPV vaccination for all persons through age 26 years. ACIP did not recommend catch-up vaccination for all adults aged 27 through 45 years, but recognized that some persons who are not adequately vaccinated might be at risk for new HPV infection and might benefit from vaccination in this age range; therefore, ACIP recommended shared clinical decision-making regarding potential HPV vaccination for these persons.

https://www.cdc.gov/mmwr/volumes/68/wr/mm6832a3.htm

Line 344-345, "Given HPV vaccination rates are relatively low nationally and internationally"

In Norway, the human papillomavirus (HPV) vaccine was introduced for girls in the childhood immunisation programme in 2009. In 2021, the coverage is up to 92 per cent for girls, which is among the highest in the world.

https://www.fhi.no/en/news/2022/high-vaccine-uptake-in-the-childhood-immunisation-programme/

In Norway, the HPV vaccine uptake increased from 72.5% in 2009 to 87.3% in 2014. The uptake increased for girls in all country background categories. Highest vaccine uptake was observed in girls with East−/South-East Asian background, 88.9% versus 82.5% in the total population.

Bjerke, R.D., Laake, I., Feiring, B. et al. Time trends in HPV vaccination according to country background: a nationwide register-based study among girls in Norway. BMC Public Health 21, 854 (2021). https://doi.org/10.1186/s12889-021-10877-8

94% of girls in Uzbekistan aged 12–14 are now covered with a first dose of human papillomavirus (HPV) vaccine, according to the country’s Ministry of Health. The HPV vaccine was first introduced into the national immunization plan in 2019, with the help of WHO and UNICEF, in order to protect girls in the country against developing cervical cancer. 

https://www.who.int/europe/news/item/07-09-2022-uzbekistan-achieves-high-hpv-vaccination-coverage-against-cervical-cancer

Public HPV vaccination programmes in Italy and Denmark were swiftly established and are among the most successful worldwide. Still, in both countries, it has been challenging to achieve and maintain the recommended coverage of > 80% in girls. In a well‐studied Italian region, vaccination coverage in girls at age 15 years (World Health Organization's gold standard) reached 76% in 2015 but decreased to 69% in 2018, likely due to work overload in public immunization centres. In Denmark, doubts about safety and efficacy of the HPV vaccine generated a decline in coverage among girls age 12–17, from 80% in 2013 down to 37% in 2015, when remedial actions made it rise again.

Bigaard J, Franceschi S. Vaccination against HPV: boosting coverage and tackling misinformation. Mol Oncol. 2021 Mar;15(3):770-778. doi: 10.1002/1878-0261.12808. Epub 2020 Oct 15. PMID: 33058497; PMCID: PMC7931130.

Incidence and mortality for cervical cancer correlate with poverty. Whilst all WHO member states report high infant measles vaccination rates, fewer than half report on HPV vaccination. Even amongst high-income countries, coverage varies widely. In upper-middle-income countries, there is a trend for higher coverage with increased health spending per capita. Four LMICs report good coverage levels, all associated with external funding. Global HPV immunisation coverage for 2018 is estimated at 12.2%

Spayne J, Hesketh T. Estimate of global human papillomavirus vaccination coverage: analysis of country-level indicators. BMJ Open. 2021 Sep 2;11(9):e052016. doi: 10.1136/bmjopen-2021-052016. PMID: 34475188; PMCID: PMC8413939.

https://pubmed.ncbi.nlm.nih.gov/34475188/

Author Response

Reviewer #1

Issanov et al. have studied the risk of HPV-related cancer in 891 HPV-vaccinated and 9,000 non-vaccinated US adults 20-59 years of age. There were four HPV-related cancers (0.4%) among the vaccinated and 58 cases among the non-vaccinated (0.8%). The difference was not statistical significant.

Comment #1

Cervical cancer is most frequently diagnosed in women between the ages of 35 and 44 with the average age at diagnosis being 50. It rarely develops in women younger than 25. In the US, the incidence rate was 7.7 per 100,000 women per year in 2014-2018. You need more power than 891 HPV-vaccinated and 9,000 unvaccinated women 20-59 years old between 2011 and 2018 to see any statistically significant results. In the manuscript, about 50% of the participants were males, without any risk of cervical cancer. 

https://cancerstatisticscenter.cancer.org

Authors’ response:

We thank the Review for this comment. Indeed, our study did not have sufficient statistical power to detect statistically significant difference. We added this limitation in the main text and provided a supplementary table with statistical power calculations. We performed two statistical power calculations based on two scenarios: 1) what is the statistical power when the outcome is cervical cancer, given the sample size? 2) what is the statistical power when the outcome is HPV-related cancers, given the sample size? (see below):

“Lastly, we also acknowledge that the present study had insufficient statistical power to detect a statistically significant difference between the vaccinated and non-vaccinated groups (Table S3). Future studies with large sample sizes are warranted to assess a statistically significant reduction.”

Table S2. Statistical power calculations based on the study sample size from the U.S. adults aged 20-59 in the National Health and Nutrition Examination Survey (NHANES), 2011-2018.

Cancer type

Number of not vaccinated

Prevalence among not vaccinateda

Number of vaccinated

Prevalence among vaccinatedb

Statistical powerc

Cervical cancer

4,259

0.003

687

0

30.1%

HPV-related cancers*

9,000

0.0063

900

0

66.1%

a Cancer prevalence estimates were calculated using data from the National Cancer Institute https://seer.cancer.gov/statfacts/

b We assumed that the vaccine would be highly effective (100%) against HPV-related cancers, so no cases would be detected in the HPV-vaccinated group.

c We calculated statistical power under two conditions: 1) when the outcome is only cervical cancer; 2) when the outcome is HPV-related cancers. The statistical power estimates were calculated using the Epi Info online calculator available from https://www.openepi.com/SampleSize/SSCohort.htm

Included only female participants

* HPV-related cancers including oropharyngeal (cancers of lip, oral cavity, pharynx, and larynx) and anogenital (cervical, vaginal, vulvar, penile) cancers.

Comment #2

  1. Usually, cervical cancer and other HPV-related cancers develops over several years. It takes at least 10 years from HPV-infection to development of invasive cancer. 

It has been estimated that the mean calculated time from infection to diagnosis of CIN3+ is 9.4 years (SD 4.1 years) (Depuydt 2012) and progression from CIN3 to invasive cervical cancer takes 10–20 years, depending on genotype (Tjalma 2013). A modelling study by Burger et al. found that the median time from acquisition of HPV to cancer detection ranged from 17.5 years to 26.0 years (Burger 2020). It is not expected to find a significant reduction in the risk of HPV-related cancer the first 10-15 years after vaccination.

  1. Tjalma WA, Fiander A, Reich O, Powell N, Nowakowski AM, Kirschner B, et al. Differences in human papillomavirus type distribution in high-grade cervical intraepithelial neoplasia and invasive cervical cancer in Europe. Int J Cancer. 2013;132(4):854–67. https://doi.org/10.1002/ijc.27713.
  2. Depuydt CE, Criel AM, Benoy IH, Arbyn M, Vereecken AJ, Bogers JJ. Changes in type-specific human papillomavirus load predict progression to cervical cancer. J Cell Mol Med. 2012;16(12):3096–104. https://doi.org/10.1111/j.1582-4934.2012.01631.x.
  3. Burger EA, de Kok I, Groene E, et al. Estimating the natural history of cervical carcinogenesis using simulation models: a CISNET comparative analysis. J Natl Cancer Inst. 2020;112(9):955–63. https://doi.org/10.1093/jnci/djz227.

Authors’ response:

Yes, we agree with the Reviewer, and we acknowledged this limitation in the discussion and added the references (see below):

“Additionally, the study timing and the observed rare events were the major limitations to finding a statistically significant association. In addition to a small number of HPV-related cancers (n=72), most vaccinated participants were between age of 20-39 (92.5%). Since HPV-related cancer development may take 10-20 years after first contracting HPV, the majority of participants were not yet at risk of developing cancers, reducing the probability of detecting the association [66-68].”

Comment #3

  1. The Advisory Committee on Immunization Practices (ACIP)* routinely recommends HPV vaccination at age 11 or 12 years. Because HPV acquisition generally occurs soon after first sexual activity, vaccine effectiveness will be lower in older age groups because of priorinfections. Exposure to HPV decreases among older age groups.

Issanov et al. studied US adults 20-59 years old. Only of few of these have been vaccinated before sexual onset.

Authors’ response:

We thank the Reviewer for this comment. 715 out of 891 vaccinated participants reported their HPV-vaccination age. Most participants were vaccinated before age 26 (13% before 15 years, 68.5% between 16-26 years old). We tried to link their vaccination age with their sexual onset responses. However, there were too few responses related to sexual activity in the NHANES database, so we could not meaningfully determine whether most participants were vaccinated before the first sexual intercourse or not.

We added the following sentence to indicate this limitation:

“As HPV vaccination before the first sexual onset, at age 11-12, is generally recommended, the vaccine might have a lower effectiveness in the older individuals, given they might have already contracted the infection. We were not able to determine whether participants were vaccinated before the first sexual onset or not, since only a few participants responded to questions about sexual behavior questions.”

Comment #4

  1. Issanov et al. have sited two studies showing reduction in HPV-related cancer among vaccinated women. Issanov et al. should also include more studies.

In a study from England, the estimated relative reduction in cervical cancer rates by age at vaccine offer were 34% (95% CI 25–41) for age 16–18 years (school year 12–13), 62% (52–71) for age 14–16 years (school year 10–11), and 87% (72–94) for age 12–13 years (school year 8), compared with the reference unvaccinated cohort. 

Falcaro M, Castañon A, Ndlela B, Checchi M, Soldan K, Lopez-Bernal J, Elliss-Brookes L, Sasieni P. The effects of the national HPV vaccination programme in England, UK, on cervical cancer and grade 3 cervical intraepithelial neoplasia incidence: a register-based observational study. Lancet. 2021 Dec 4;398(10316):2084-2092. doi: 10.1016/S0140-6736(21)02178-4. Epub 2021 Nov 3. PMID: 34741816.

https://pubmed.ncbi.nlm.nih.gov/34741816/

In a study from Denmark including 867 689 women aged 17-30 years living in Denmark October 2006-December 2019, 36.3% were vaccinated at age 16 years and younger, and during follow-up, 19.3% and 2.3% were vaccinated at ages 17-19 years and 20-30 years, respectively. For women vaccinated at ages 16 years and younger or 17-19 years, the IRRs of cervical cancer were 0.14 (95% CI = 0.04 to 0.53) and 0.32 (95% CI = 0.08 to 1.28), respectively, compared with unvaccinated women.

Kjaer SK, Dehlendorff C, Belmonte F, Baandrup L. Real-World Effectiveness of Human Papillomavirus Vaccination Against Cervical Cancer. J Natl Cancer Inst. 2021 Oct 1;113(10):1329-1335. doi: 10.1093/jnci/djab080. PMID: 33876216; PMCID: PMC8486335.

https://pubmed.ncbi.nlm.nih.gov/33876216/

Recently, the International Papillomavirus Society (IPVS) issued a ‘call to action’ to health authorities to achieve elimination of cervical cancer as a public health problem. 1 In principle, cervical cancer rates can eventually be reduced to near-zero given the highly effective primary prevention (HPV vaccination) and secondary prevention (cervical screening) interventions now available.

Australia is poised to be the first country to approach cervical cancer elimination, since it has now fully implemented all these major prevention interventions. Australia was the first country to introduce a national publicly-funded HPV vaccination program in 2007, with a wide catch-up age range from 12 to 26 years. In 2013, Australia introduced vaccination for adolescent males, and in 2018 the next generation nonavalent vaccine was introduced, which protects against seven carcinogenic types which are associated with ~90% of cervical cancers. Multiple studies have documented the impact on health outcomes: the prevalence of vaccine-included type-specific infections in young women aged 25-35 years has now dropped by a factor of 10 (even in unvaccinated females, due to herd immunity), 5 the prevalence of anogenital warts has dropped substantially in both females and heterosexual males, 6 cervical precancerous abnormalities (CIN2/3) have now dropped by 41% nationally in women aged 2024 years, 7 and the rate of excisional treatment has now also fallen in young women. 8

In the intermediate term, cervical cancer rates are expected to halve (again) by 2035, and mortality rates should remain stable until about 2020, but then decline by 45% by 2035. These findings indicated that both HPV vaccination and primary HPV screening represent significant and timely steps in Australia’s journey towards elimination of cervical cancer.

https://www.hpvworld.com/articles/australia-on-track-to-be-the-first-country-to-achieve-cervical-cancer-elimination/

Authors’ response:

We appreciate the recommended references. We expanded the discussion and added the references (see below):

“Similarly, a study from England reported relative incidence rate reductions in cervical cancer among women who were offered the vaccine as follows: between age 16-18 years, 34% (95% CI 25%-41%); between age 14-16 years, 62% (95% CI 52%-71%); and between age 12-13 years, 87% (95% CI 72%-94%), in comparison to unvaccinated women[48]. Another study from Denmark found a statistically significant reduction in incidence rate of cervical cancer in women who were vaccinated at age 16 years or younger – 86% (95% CI 47%; 96%) – while failing to find any reduction at ages 17 years or older[49]. Lastly, the compre-hensive HPV vaccination program in Australia is expected to halve cervical cancer rates by 2035[50]..”

Minor revisions

Comment #5

Line 40, "High-risk HPV types – HPV 6, 11, 16, 18 and other types – are causally associated with 90% of cervical" => "Seven high-risk HPV types (16, 18, 31, 33, 45, 52 and 58) – are causally associated with 90% of cervical"

(HPV type 6 and 11 are low-risk HPV-types causing genital warts and not HPV-related cancers. HPV type 16 and 18 are causing 70% of all cases of cervical cancer. The 9-valent HPV-vaccine covers the 7 hrHPV-types causing 90% of all cases of cervical cancer).

Authors’ response:

We thank the Reviewer for this comment. We made changes accordingly in the text.

Comment #6

Line 44-45, "There are highly efficacious HPV vaccines against the high-risk HPV infection types, recommended at the age of 9-14 for girls and boys"

Females between 9 and 45 years of age can be vaccinated with Cervarix, Gardasil or Gardasil 9 to prevent cervical cancer and precancerous cervical changes. Gardasil and Gardasil 9 may also prevent vaginal, vulvar and anal cancers and their precancers, as well as anogenital warts. 

In Canada, about one-third of HPV-related cancers occur in males. Gardasil and Gardasil 9 are available for boys and young men between the ages of 9 and 26 to prevent anal cancer, its precancer and anogenital warts.

https://cancer.ca/en/cancer-information/reduce-your-risk/get-vaccinated/human-papillomavirus-hpv

The Advisory Committee on Immunization Practices (ACIP)* routinely recommends HPV vaccination at age 11 or 12 years; vaccination can be given starting at age 9 years. Catch-up vaccination has been recommended since 2006 for females through age 26 years, and since 2011 for males through age 21 years and certain special populations through age 26 years. This report updates ACIP catch-up HPV vaccination recommendations and guidance published in 2014, 2015, and 2016 (13). Routine recommendations for vaccination of adolescents have not changed. In June 2019, ACIP recommended catch-up HPV vaccination for all persons through age 26 years. ACIP did not recommend catch-up vaccination for all adults aged 27 through 45 years, but recognized that some persons who are not adequately vaccinated might be at risk for new HPV infection and might benefit from vaccination in this age range; therefore, ACIP recommended shared clinical decision-making regarding potential HPV vaccination for these persons.

https://www.cdc.gov/mmwr/volumes/68/wr/mm6832a3.htm

Authors’ response:

We appreciate the comment and added additional information in the introduction.

“There are highly efficacious HPV vaccines against the high-risk HPV infection types, recommended at the age of 9-14 for girls and boys[10]. The Advisory Committee on Immunization Practices also recommends catch-up HPV vaccination for all individuals up to age 26 years old while HPV vaccination for adults aged 27-45 should be based on shared clinical decision-making[12,13].”

Comment #7

Line 344-345, "Given HPV vaccination rates are relatively low nationally and internationally"

In Norway, the human papillomavirus (HPV) vaccine was introduced for girls in the childhood immunisation programme in 2009. In 2021, the coverage is up to 92 per cent for girls, which is among the highest in the world.

https://www.fhi.no/en/news/2022/high-vaccine-uptake-in-the-childhood-immunisation-programme/

In Norway, the HPV vaccine uptake increased from 72.5% in 2009 to 87.3% in 2014. The uptake increased for girls in all country background categories. Highest vaccine uptake was observed in girls with East−/South-East Asian background, 88.9% versus 82.5% in the total population.

Bjerke, R.D., Laake, I., Feiring, B. et al. Time trends in HPV vaccination according to country background: a nationwide register-based study among girls in Norway. BMC Public Health 21, 854 (2021). https://doi.org/10.1186/s12889-021-10877-8

94% of girls in Uzbekistan aged 12–14 are now covered with a first dose of human papillomavirus (HPV) vaccine, according to the country’s Ministry of Health. The HPV vaccine was first introduced into the national immunization plan in 2019, with the help of WHO and UNICEF, in order to protect girls in the country against developing cervical cancer. 

https://www.who.int/europe/news/item/07-09-2022-uzbekistan-achieves-high-hpv-vaccination-coverage-against-cervical-cancer

Public HPV vaccination programmes in Italy and Denmark were swiftly established and are among the most successful worldwide. Still, in both countries, it has been challenging to achieve and maintain the recommended coverage of > 80% in girls. In a well‐studied Italian region, vaccination coverage in girls at age 15 years (World Health Organization's gold standard) reached 76% in 2015 but decreased to 69% in 2018, likely due to work overload in public immunization centres. In Denmark, doubts about safety and efficacy of the HPV vaccine generated a decline in coverage among girls age 12–17, from 80% in 2013 down to 37% in 2015, when remedial actions made it rise again.

Bigaard J, Franceschi S. Vaccination against HPV: boosting coverage and tackling misinformation. Mol Oncol. 2021 Mar;15(3):770-778. doi: 10.1002/1878-0261.12808. Epub 2020 Oct 15. PMID: 33058497; PMCID: PMC7931130.

Incidence and mortality for cervical cancer correlate with poverty. Whilst all WHO member states report high infant measles vaccination rates, fewer than half report on HPV vaccination. Even amongst high-income countries, coverage varies widely. In upper-middle-income countries, there is a trend for higher coverage with increased health spending per capita. Four LMICs report good coverage levels, all associated with external funding. Global HPV immunisation coverage for 2018 is estimated at 12.2%

Spayne J, Hesketh T. Estimate of global human papillomavirus vaccination coverage: analysis of country-level indicators. BMJ Open. 2021 Sep 2;11(9):e052016. doi: 10.1136/bmjopen-2021-052016. PMID: 34475188; PMCID: PMC8413939.

https://pubmed.ncbi.nlm.nih.gov/34475188/

Authors’ response:

We really appreciate the provided text extracts and references. We revised the text accordingly adding these references (see below):

“While several high-income countries (e.g., Norway, Italy and Denmark)[59-61] and a few middle-income countries (e.g., Uzbekistan)[62] have reached and maintained high HPV vaccination coverage, the U.S. is experiencing challenges to maintain an adequate vaccination rate[30,63]. On the other hand, only less than a quarter of low-income and less than third of lower-middle-income countries had introduced the HPV vaccination [64]. Given HPV vaccination rates are relatively low nationally[27,55] and internationally [56] and overall hesitation to introduce HPV vaccine in low- and middle-income countries [64], study findings can be used by policymakers and health professionals to further encourage HPV vaccination uptake among adolescents aged 11-15 and to support decision-making about HPV vaccination among adults eligible to be vaccinated and parents considering immunizing their children.”

Reviewer 2 Report

This study is interesting, it is very clear and well described. Vaccination coverage rate is incredibly low compared to many European countries. Also it is surprising to find no statistically significant aOR between vaccination and HPV cancer related. Several studies (Sweden, Switzerland, Australia, Scotland, etc.) show a reduction in HPV infection and cancer following vaccination.

The main weakness of the study is that the variables are obtained by self-reporting which may be an important response bias, but this is well discussed in the study.

I have no comments other than to put the country of the study in the title.

Congratulations to the authors for their study.

Author Response

Reviewer #2

This study is interesting, it is very clear and well described. Vaccination coverage rate is incredibly low compared to many European countries. Also it is surprising to find no statistically significant aOR between vaccination and HPV cancer related. Several studies (Sweden, Switzerland, Australia, Scotland, etc.) show a reduction in HPV infection and cancer following vaccination.

Comment #8

The main weakness of the study is that the variables are obtained by self-reporting which may be an important response bias, but this is well discussed in the study.

I have no comments other than to put the country of the study in the title.

Congratulations to the authors for their study.

Authors’ response:

We thank the Reviewer for time spent reviewing our manuscript. We added the country name in the study title.

Reviewer 3 Report

The paper is well written with enough information.

Introduction:

-        The authors reported two studies that also looked into the link between HPV vaccination and the cancer chances. One study reported that the chances become small for vaccinated people. This study also claims the same for US population. Then I would suggest to mention “for US population” in the title.

References:

-        Many references are at least 7 years old. Please update them with more recent articles.

Author Response

Reviewer #3

Comment #9:

The paper is well written with enough information.

 Introduction:

The authors reported two studies that also looked into the link between HPV vaccination and the cancer chances. One study reported that the chances become small for vaccinated people. This study also claims the same for US population. Then I would suggest to mention “for US population” in the title.

 Authors’ response:

We thank the Reviewer for the comment. We added the country name in the study title.

Comment #10:

References:

Many references are at least 7 years old. Please update them with more recent articles.

Authors’ response:

We thank the Reviewer for the comment. We have updated some references (#6, #48, #49, #50, #68) to more recent publications (2020-2021).